# Effects of Tangerine Essential Oil on Brain Waves, Moods, and Sleep Onset Latency

**DOI:** 10.3390/molecules25204865

**Published:** 2020-10-21

**Authors:** Supaya Chandharakool, Phanit Koomhin, Jennarong Sinlapasorn, Sarunnat Suanjan, Jantamas Phungsai, Noppharat Suttipromma, Sumethee Songsamoe, Narumol Matan, Apsorn Sattayakhom

**Affiliations:** 1School of Allied Health Sciences, Walailak University, Nakhonsithammarat 80160, Thailand; supaya.ch@mail.wu.ac.th (S.C.); drank.abc@gmail.com (J.S.); sarunnat.suan@gmail.com (S.S.); jantamas.ph@gmail.com (J.P.); mike18102540@gmail.com (N.S.); 2School of Medicine, Walailak University, Nakhonsithammarat 80160, Thailand; phanit.ko@mail.wu.ac.th; 3Center of Excellence in Innovation on Essential Oil, Walailak University, Nakhonsithammarat 80160, Thailand; sumethee5192@gmail.com (S.S.); nnarumol6296@gmail.com (N.M.); 4Research Group in Applied, Computational and Theoretical Science (ACTS), Walailak University, Nakhonsithammarat 80160, Thailand; 5School of Agricultural Technology, Walailak University, Nakhonsithammarat 80160, Thailand

**Keywords:** *Citrus tangerina*, tangerine, limonene, electroencephalography activity, sleep onset latency

## Abstract

Tangerine (*Citrus tangerina*) is one of the most important crops of Thailand with a total harvest that exceeds 100,000 tons. Citrus essential oils are widely used as aromatherapy and medicinal agents. The effect of tangerine essential oil on human brain waves and sleep activity has not been reported. In the present study, we therefore evaluated these effects of tangerine essential oil by measurement of electroencephalography (EEG) activity with 32 channel platforms according to the international 10–20 system in 10 male and 10 female subjects. Then the sleep onset latency was studied to further confirm the effect on sleep activity. The results revealed that different concentrations, subthreshold to suprathreshold, of tangerine oil gave different brain responses. Undiluted tangerine oil inhalation reduced slow and fast alpha wave powers and elevated low and mid beta wave powers. The subthreshold and threshold dilution showed the opposite effect to the brain compared with suprathreshold concentration. Inhalation of threshold concentration showed effectively decreased alpha and beta wave powers and increased theta wave power, which emphasize its sedative effect. The reduction of sleep onset latency was confirmed with the implementation of the observed sedative effect of tangerine oil.

## 1. Introduction

Nowadays, the trend of organic products and natural consumers is growing in agricultural product marketing in the world. This may therefore be an option for people to select natural products more than in the past [1]. Its effects turn the niche market to the mass market, and one of the famous natural products is essential oils. In 2018, the global essential oils market demand was reported as 226.9 kilotons, and the market forecast is that it is likely to increase steadily [2]. Essential oil products are extracted from several parts of a plant, such as flowers, leaves, roots, and fruits, which contain different chemical-like hydrocarbons, alcohol, aldehydes, esters, ethers, ketones, oxides phenols, and terpenes. Different essential oils have different effects on the human nervous system such as the relaxing effect of lavender oil, the vigilance effect of jasmine oil and rosemary oil, and the attention effect of rose oil and orchid oil on the brain and cognitive functions [3,4,5]. Moreover, the analysis of EEG revealed that lavender oil increased the power of theta and alpha wave [6], and rosemary essential oil can also decrease alpha1 and alpha2 waves and increase beta waves [7]. The effect of essential oil begins during inhalation of the essential oil molecules through the nostril and attachment to odorant receptors, activating G protein-coupled receptor. It will then activate synaptic transmission to the central nervous system [8]. The signal input finally ascends to several brain regions that are responsible for olfactory perception, autonomic homeostasis, and other higher brain functions [9]. There are many recording or imaging methods measuring the effect of stimuli on brain. Electroencephalography (EEG) is one of the most popular methods for observing brain oscillations due to its cost efficiency. Brain waves are classified by frequencies ranging from 0.05 to 500 Hz such as delta wave (0–4 Hz), theta wave (4–8 Hz), alpha wave (8–13 Hz), beta wave(13–30 Hz), and gamma (above 30 Hz) [10]. In this study, we focused on tangerine *(Citrus tangerina)* peel essential oil. According to the reports of the office of agricultural economics of Thailand in 2016, tangerine is Thailand’s most important crop with high productivity of about 146,526 tons. Tangerine peel that is rich in essential oil is usually treated as waste or by-product. The previous study showed that tangerine peel extract has significant antioxidant activity, which can possibly be applied to food and medicine [11]. The components of tangerine essential oil consist of limonene, γ-terpinene, paracymene, β-myrcene, β-pinene, α-pinene, and others. The content of each component may vary by geographic location. Limonene is major component of tangerine essential oil. Many studies revealed that limonene has effects on the human brain and body responses including an increase of systolic blood pressure, alertness level and sleeplessness [12,13]. However, there is no evidence to support tangerine oils effects and an effective dose with designed brain waves and responses. Therefore, this study continued investigating an implemented illustration of obtained knowledge. Sleep latency was then studied to further confirm the effect of the selected essential oil dilution.

## 2. Results

### 2.1. Effects of Tangerine Essential Oil Inhalation on Brain Responses

Tangerine essential oil was analyzed using GC-MS technique. The GC-MS analysis of the tangerine oil identified the compounds presented in Figure 1. The major compounds of this tangerine oil are d-Limonene (74.47%) and γ-Terpinene (10.86%). Other compounds were α-Phellandrene, Camphene, β-Pinene, Sabinene, 3-Carene, β-Myrcene, 2,4,6-Octatriene, 3,4-dimethyl-, o-Cymene, Octanal, Nonanal, and linalool (Table 1). Brain wave alteration was analyzed for each representative frequency of each frequency, slow alpha wave (9 Hz), fast alpha wave (12 Hz), low beta wave (14 Hz), mid beta wave (17 Hz), and high beta wave (25 Hz). Power spectral analysis technique was used to clearly identify the differences. All of electrodes were analyzed to investigate overall activity of the brain. Volunteers were the same age and had enough sleeping hours during the night before the day of the experiment. Mixed gender, male, and female data were divided to see the gender difference. Tangerine essential oil inhalation reduced slow and fast alpha wave powers. Low and mid beta wave power increased during oil inhalation. High beta wave power showed no difference among groups. For female subjects, the oil inhalation reduced slow alpha wave power but the oil increased low and mid beta wave powers. The oil inhalation showed slight effects in the male subjects, which only reduced fast alpha wave power (Figure 2). Pz, AF4, and T7 electrodes showed a significant increase of slow alpha wave power, a significant decrease of fast alpha wave power, and a significant increase of low beta wave power, as shown in Table 2. Tangerine essential oil inhalation showed gender difference. We did not directly analyze gender difference between female and male, however, female responses to the oil can be obviously observed in the frequency of alpha and beta wave powers.

### 2.2. Effects of Tangerine Essential Oil on Brain Regions

To analyze the brain activities of each dedicated region, we divided recording electrodes into four regions according to the responsible cortex underneath the skull. It consists of a frontal region, centro-temporal region, parietal region, and occipital region. Single channel analysis suggested that the tangerine oil can produce a significant change in three different electrode sites. To specifically study activities that contributed from each brain region, we defined each responsible electrode group for each brain region. The frontal region is composed of Fp1, Fp2, AF3, AF4, F7, F3, FZ, F4, FC5, FC1, FC2, and FC6, which record cortical activity in the frontal area of the brain. Centro-temporal region is composed of T7, C3, CZ, C4, T8, CP5, CP1, CP2, and CP6, which are responsible for virtually the middle and temporal area. The parietal region is composed of P7, P3, PZ, P4, and P8, which are responsible for parietal region. Finally, the occipital region is composed of PO7, PO3, OZ, PO4, and PO8, which are responsible for the occipital region. There was an increase of slow alpha wave power in the frontal region and a decrease of slow alpha wave power in centro-temporal, parietal, and occipital regions in mixed gender (Figure 3). No difference was observed in male subjects for slow wave power. In female subjects, the result showed a reduction of brain wave power in centro-temporal and parietal regions. For fast alpha wave power, the result showed a reduction of brain wave power in the frontal region in mixed gender and showed both a reduction of brain wave power in both genders in the frontal region (Figure 4). For low beta wave power, the result showed an increase of brain wave power in frontal, centro-temporal, and parietal regions (Figure 5). Only female subjects showed increases of this brain wave power in frontal, centro-temporal, and parietal regions. For mid beta wave power, it showed an increase of brain wave power in the frontal region and only in female subjects was an increase in frontal region observed (Figure 6). For high beta wave power, the result showed an increase of brain wave power in the centro-temporal region, which was observed in male subjects. This brain wave power was observed as an increase in occipital region in female subjects (Figure 7).

### 2.3. Effects of Subthreshold and Threshold Oil Dilution on Brain Wave, Sleep Onset Latency, and Emotions

We tested subthreshold and threshold dilutions of tangerine oil in female subject because of a sensitivity issue. Subthreshold and threshold were examined by two dilution tests that are 1:8000 and 1:1000 dilutions, respectively. Interestingly, subthreshold and threshold dilutions caused different brain responses compared with suprathreshold concentration (undiluted concentration). The result showed slightly increased theta wave power and slightly decreased alpha wave power. No beta wave power reduction was observed during 1:8000 dilution inhalation. It revealed more reduction of alpha and beta wave powers and an increase of theta wave power in 1:1000 dilution (Figure 3). The subthreshold and threshold dilution showed opposite effects on the brain compared with suprathreshold concentration, which emphasizes the sedative effect of the oil on these levels. Tangerine oil was then further tested for sleep onset latency. The result showed that the oil with 1:1000 dilution significantly reduced sleep onset latency in female subjects compared with baseline sleep onset latency (Figure 8). Emotional states were self-estimated by subjects. The results showed that suprathreshold concentration tends to increase the fresh feeling in both female and male subjects (Table 3). This study focused on five emotions consisting of good, active, drowsy, fresh, and romantic. Romantic feeling was the lowest, among others.

## 3. Discussion

The tangerine essential oil caused statistically significant effects to the brain in different brain regions. It increased brain-wave sub band of beta wave power but decreased the alpha wave power [10]. This study shows the vigilance effect of undiluted tangerine oil. Beta wave power is observed when awake, especially active thinking, focused, high alert for mid beta wave power and fast idle and musing for low beta wave power [14]. Alpha wave power is observed during several states of mind or activities. Slow alpha wave power correlated with relaxed state, eyes close condition, calm state, and resting state. Fast alpha wave power correlated with idling state. The reduction of this brain wave power suggests a brain functional shift to a more alert state of mind of beta [6]. Limonene is a major component in tangerine oil detected by GC-MS. Another study found that autonomic nervous system was increased by (+)-limonene, which was observed by the increase of sympathetic activity parameters [13]. Sowndhararajan et al. also reported beta wave power increase by (+)-limonene inhalation [15]. The increase of alertness by tangerine oil or limonene might be because of several underlying brain processes and the psychological basis. Brainstem–thalamo–cortical pathways, suprachiasmatic nucleus—circadian rhythm, hypothalamo–pituitary–adrenal axis and limbic system, as well as the metabolic system and substrates need to be further investigated to explain the underlying mechanisms of these volatile fragrances [16]. The vigilance effect of tangerine oil is similar to rosemary oil. Previous study suggested that volatile organic in *Chrysanthemum indicum*, 1,8-cineole, can increase the low beta at several sites, excluding frontal, parietal and occipital. By contrast, only frontal is detected in the present study [17]. In the case of mid beta wave of 15–20 Hz with high significance in active thinking and high alertness, Kim M. has shown that mid beta wave was increased during inhalation of (+)-α-pinene and terpinolene, which is correlated to decreased mid beta, indicating relaxation and reducing stress [15,18]. In addition, the effect of essential oil of *Inula helenium* root decreases the mid beta but only in the parietal region [19]. Each region of the brain is responsible for different functions from homeostatic functions to higher functions. In this study, we divided electrodes into four groups representing frontal, centro-temporal, parietal, and occipital regions. The frontal area is responsible for many important cognitive functions such as working memory, calculation, motivation, and executive function. This brain region collaborates with other brain areas in other deep nuclei to do these elaborative functions. The alteration of brain activities in this brain area may correlate with the mentioned duties. The temporal lobe and central area of the brain are responsible for limbic system-related function, motor function and sensory function. The parietal region is responsible for associative function, and the occipital lobe is responsible for vision. Previous studies showed increased slow alpha in different regions. Neroli and grapefruit oils influenced activity in the occipital region, which indicates relaxation state and helps to reduce cortical deactivation [20]. The mixing of bergamot and lavender leads to the increase of relative fast and slow alpha waves in the prefrontal region, which is also helping to relax [21]. Orchid oil increased the parietal region activity [22]. Low beta is decreased during inhalation of (+)-α-pinene, which is specific in male subjects. The increase in beta wave has also been correlated with an enhanced drowsy state [17]. The decrease of fast alpha wave power was observed during resting and concentration period of mind. The other studies also showed the different results. When the subjects were exposure to agarwood incense, the fast alpha wave was significantly increased and increased during inhalation of (+)-α-pinene and (+)-β-pinene [17,20]. Moreover, the present study showed an emotional state after inhalation of the undiluted oil, which happens in the same way as brain wave power. The visual analog score of orchid oil inhalation also showed higher scores in elegance and freshness [18].

In this study, we did not investigate the related functions of each brain area; however, we investigate a function related with arousal level. The sleep onset-latency was measured in this study to further extend the knowledge related with sedative function of tangerine oil. We prepared the oil in the level of subthreshold and threshold concentration instead of suprathreshold, undiluted oil. We found that both the subthreshold and threshold showed an interesting function opposite to undiluted oil. It showed a sedative effect in these ranges of dilutions. We hypothesized that the oil may help sleep function. Only female subjects were recruited in this study and the result confirms the hypothesis. The result briefly showed the effect on the alertness system. Sleep onset latency decreased in female subjects. *Lavandula angustifolia* essential oil also decreased sleep onset latency, which is like the threshold level of tangerine oil [23]. In vivo study revealed that limonene significantly increased γ-aminobutyric acid (GABA) and other neurotransmitter changes [24]. GABA is acting though GABA_A_ receptor, which is potentiating chloride current resembling alcohol, barbiturates, and benzodiazepines. Moreover, other neurotransmitters were reported to change after limonene administration such as dopamine (DA), serotonin (5-HT), and glutamate (Glu). These neurotransmitters play an important role related with the waking system of the brain so the direct effect of the oil to neurotransmitter function may be the underlying mechanism related with its sedative and vigilance effects. This detailed mechanism is still needed to be investigated. The gender difference was indirectly observed in this study. Previous studies also showed different brain waves between female and male during inhalation of (+)-α-pinene and (+)-β-pinene as well as black pepper essential oil [17]. Moreover, the other studies revealed that reproductive hormones and odor reception are the cause of different responses between genders [25]. Different olfactory sensitivity between genders of many animal species may be caused by the physical receptor [26]. In other studies, corpus collosum of females is larger than males; in addition, the development of cerebral lateralization of females is also faster than males [27,28]. Furthermore, the other study showed that females responded to the molecules of the essential oil more than male did, and Haehner et al. also revealed that responses of females and males to grapefruit or a combination of orange, lime and lemon fragrances are different [29,30]. In addition, sex-related performance can be explained by reproductive hormone influences on odor perception. Doty et al. explained that alterations of olfactory performance were observed in different states of hormone reproduction, such as gonadectomy, hormone replacement therapy, pregnancy, and menstrual cycle [31]. All the above are consistent with the present study because females have more brain activity during inhalation of tangerine essential oil compared with male subjects. Cognitive tasks should be further examined in the future to study the related brain functions. A previous study showed that San-Jo-In essential oil can increase attention and relaxation with a significant increase in fast alpha power at the left prefrontal, right prefrontal and left frontal regions in the same way as the effect of lavender and bergamot [32]. Moreover, alpha increase in the frontal brain region was related to moods, helping to decrease tension and stress states [6]. With regards to the effect of increased beta wave power in frontal regions, other studies showed that it was related to alertness and concentration state including memory task performance improvement [33,34]. Kim et al. showed that alpha wave was increased during inhalation of (+)-α-pinene and (+)-β-pinene, which may improve brain functions and enhance relaxation. This study only measured overall brain responses and related functions such as sleep function. The mechanism related with brain communication is still obscured during essential oil inhalations.

## 4. Materials and Methods 

### 4.1. Subject

The study protocols of this study were approved by the Institutional Review Board, Walailak University, and all the subjects were informed according to the standard of ethical practice in research involving humans. The approval numbers are WUEC-16-043-01 (14 June 2017) and WUEC-19-140-01 (11 September 2019). A total of 20 healthy subjects, consisting of 10 Thai males and 10 Thai females aged 19–25 years old were recruited for tangerine oil evaluation. Ten female subjects were recruited to examine sleep onset latency. All subjects were right-handed to limit dominant activity interferences and were undergraduate university students. All subjects had adequate sleep with at least 6 h of sleep prior to the day of experiment. All female subjects declared not to be on a menstrual period on the day of experiment. Alcohol consumption, medications and smoking were prohibited for a week before the day of experiment. Moreover, none of the subjects experienced neurological injuries and olfactory disorder. The diet pattern of the subjects was the traditional Thai eating pattern [35]. The subjects were requested to confirm an olfactory evaluation test using the smell test. Subjects’ baseline characteristics are shown in Table 4; Table 5. The method was slighty modified from Koomhin et al. 2020 [36].

### 4.2. Tangerine Oil Preparation and Administration

The Tangerine oil extracted using steam distillation was obtained from Thai China Flavours & Fragrances Co., Ltd.(Bangkok, Thailand). The tangerine oil was analyzed by gas chromatography–mass spectroscopy (GC–MS) (Agilent 5977A, Agilent Technologies, Inc., Santa Clara, CA, USA) using two VF-WAXms columns (30 m × 250 µm, film thickness 0.25 μm). The oven temperature was programmed at 35–200 °C (10 °C/min), 200 °C (5 min), 200–250 °C (4 °C/min), and 250 °C (10 min); injector temperature: 250 °C; carrier gas: helium: splitting ratio 500:1; injection volume: 1 μL; interface temperature: 250 °C; MS source temperature: 230 °C; and MS quadrupole temperature: 150 °C. The compounds were identified by their retention indices and their mass spectra. The components of the oil were identified by comparison of their mass spectra fragmentation and computer matching using the Wiley 10 and NIST 14 libraries (Database/ChemStation data system). Fifty microliters of essential oil were spotted on a filter paper and the filter paper was attached to the subject’s nose at approximately 3-cm distance for analysis of the effects of tangerine essential oil inhalation on brain waves and mood responses compared to baseline (without essential oil) as shown in Figure 9. To illustrate concentration effect, different concentrations of tangerine oil including undiluted concentration, 1:1000 dilution, and 1:8000 dilution, which represent suprathreshold, threshold, and subthreshold, respectively, were used for experiment. To evaluate the effect of essential oil on sleep study, 25 μL of essential oil that contains a tangerine essential oil concentration of 1:1000 dilution was spotted on a filter paper and two sets of this filter paper were inserted into the pillow case of the subject.

### 4.3. Electroencephalography and Sleep Onset Latency

The international 10–20 system was used for an electroencephalographic study. The subjects wore Quik cap. After an electrode alignment, Quik gel was loaded to each designated electrode and the impedance was adjusted to be lower than 5 kΩ using Compumedics Neuroscan, SynAmps RT 64-channel amplifier (Compumedics Neuroscan, Victoria, Australia). Thirty-two electrodes were chosen and were divided into four categories consisting of regions frontal (Fp1, Fp2, AF3, AF4, F7, F3, FZ, F4, FC5, FC1, FC2, and FC6), centro-temporal region (T7, C3, CZ, C4, T8, CP5, CP1, CP2, and CP6), parietal region (CP5, CP1, CP2, CP6, P7, P3, PZ, P4, and P8) and occipital region (PO7, PO3, OZ, PO4, and PO8). The sampling rate was adjusted to 1 kHz. The raw data was filtered and removed eye-blink artifacts using Curry7 analysis software (Compumedics Neuroscan, Victoria, Australia). Three 10-s epochs were analyzed using fast-Fourier transform method. Power spectral analysis was then used to show the differences. For sleep staging study, 8-channel electrodes (F3, F4, C3, C4, O1, O2, A1, and A2), electromyography, and electrooculography were recorded. Sleep onset latency was analyzed by the duration from switch off the light to the presentation of sleep spindles or K-complexes. 

### 4.4. Score of Moods State Response

This study used visual analog scale for assessment of the internal mood state of the subjects after inhalation of tangerine essential oil. The internal moods of subjects were scored by themselves. Briefly, 10-centimeter horizontal lines of visual analog scale were marked according to their feeling. This visual analog scale provides an assessment of good, active, drowsy, fresh, and romantic.

### 4.5. Data Analyses and Statistics

Data were shown as mean ± SEM. Parametric data and median for non-parametric data. Analyses of variance was performed followed by Wilcoxon matched-pairs signed rank test for non-parametric data, respectively. Statistical differences were considered at *p*-value less than 0.05. In this study, data will be compared between, before and during inhalation.

Data analysis for EEG: raw data were processed by power spectral analysis, and frequency domain data are characterized as slow alpha (8–11 Hz), fast alpha (11–13 Hz), low beta (13–15 Hz), mid beta (15–20 Hz), and high beta (20–30 Hz). This describes the distribution of electrical activity across the brain regions and was produced to show overview activities using GraphPad Prism 8.

## 5. Conclusions

Tangerine essential oil has a stimulation effect on the human brain, which is observed by the reduction of slow and fast alpha wave powers and the increase of low beta and mid beta wave powers. Interestingly, alpha and beta wave power were clearly decreased during inhalation of threshold concentration of the oil, indicating the sedative effect of tangerine oil on this level. These results reveal the different effects of the different concentrations of tangerine oil, which may contribute to the concentration-dependent effect of essential oil inhalation. Moreover, the reduction of sleep onset latency was further confirmed by the sedative effect of threshold-level tangerine oil.

## Figures and Tables

**Figure 1 molecules-25-04865-f001:**
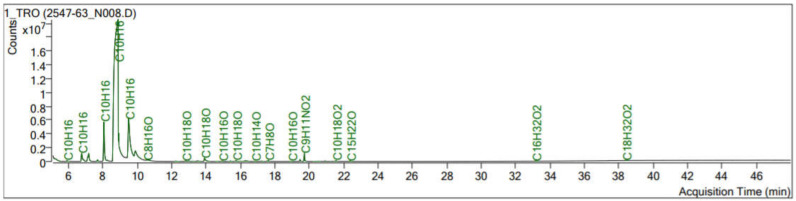
Chromatogram of tangerine essential oil analyzation using GC-MS technique.

**Figure 2 molecules-25-04865-f002:**
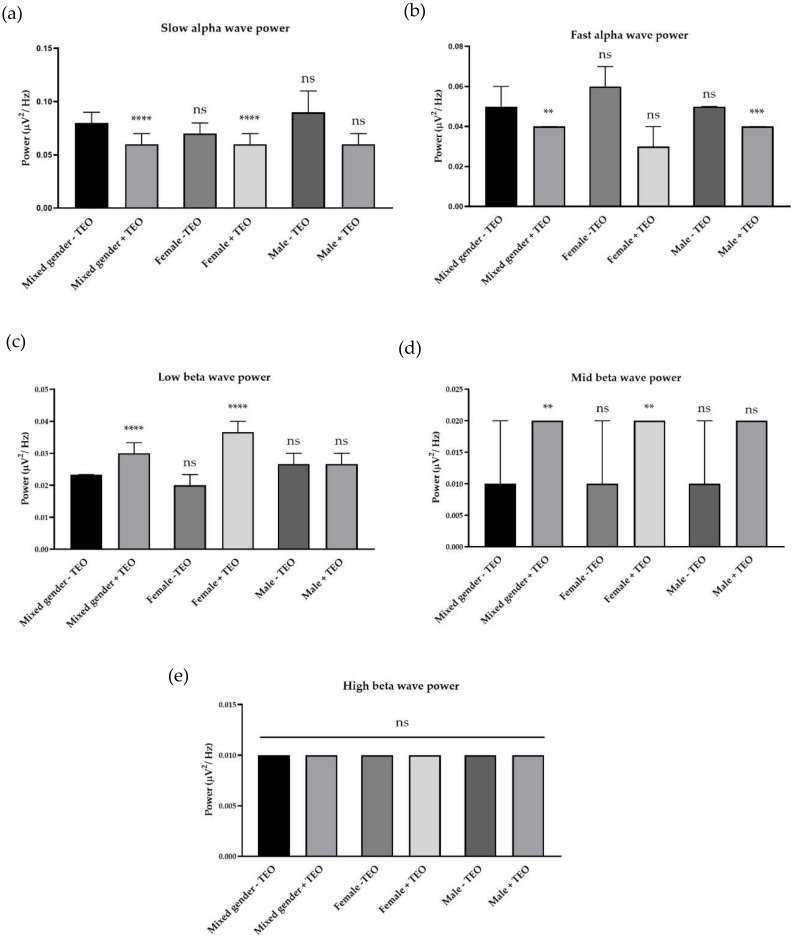
Effects of tangerine essential oil inhalation: (**a**) slow alpha (9 Hz); (**b**) fast alpha (12 Hz); (**c**) low beta (14 Hz); (**d**) mid beta (17 Hz); (**e**) high beta (25 Hz). *p* value: ns ≥ 0.05, ** <0.01, *** <0.001, **** <0.0001.

**Figure 3 molecules-25-04865-f003:**
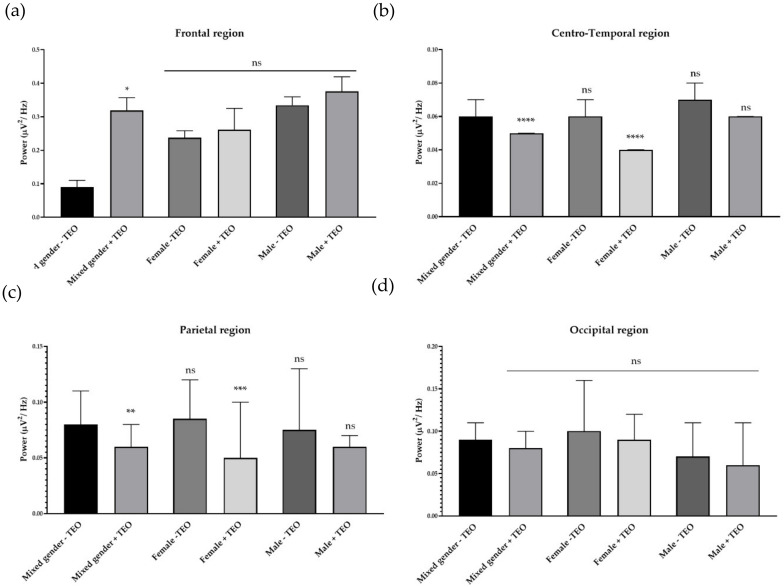
Effects of tangerine essential oil on slow alpha (9 Hz): (**a**) Frontal region; (**b**) Centro-Temporal region; (**c**) Parietal region; (**d**) Occipital region. *p* value: ns ≥ 0.05, * <0.05, ** <0.01, *** <0.001, **** <0.0001.

**Figure 4 molecules-25-04865-f004:**
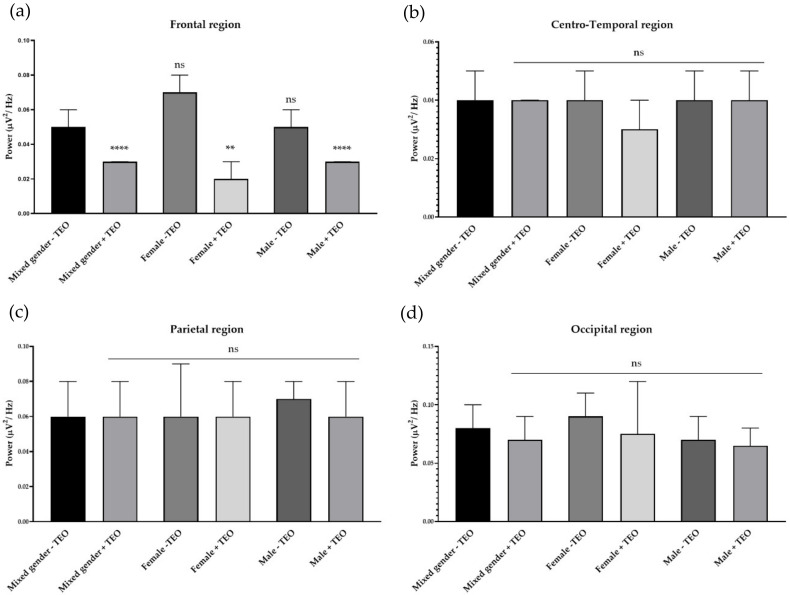
Effects of tangerine essential oil on fast alpha (12 Hz): (**a**) Frontal region; (**b**) Centro-Temporal region; (**c**) Parietal region; (**d**) Occipital region. *p* value: ns ≥ 0.05, ** <0.01, **** <0.0001.

**Figure 5 molecules-25-04865-f005:**
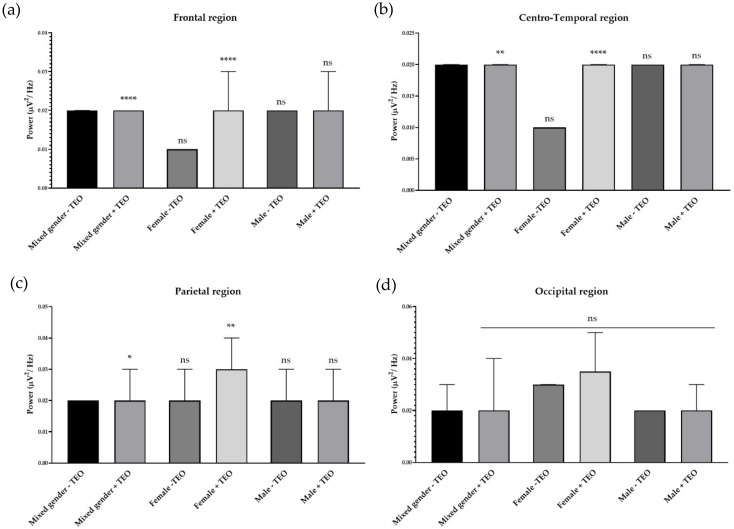
Effects of tangerine essential oil on low beta (14 Hz): (**a**) Frontal region; (**b**) Centro-Temporal region; (**c**) Parietal region; (**d**) Occipital region. *p* value: ns ≥ 0.05, * <0.05, ** <0.01, **** <0.0001.

**Figure 6 molecules-25-04865-f006:**
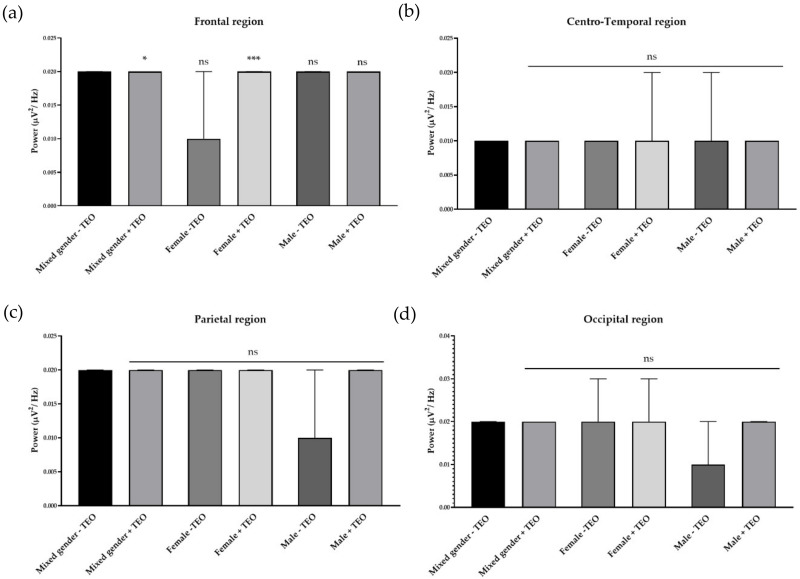
Effects of tangerine essential oil on mid beta (17 Hz): (**a**) Frontal region; (**b**) Centro-Temporal region; (**c**) Parietal region; (**d**) Occipital region. *p* value: ns ≥ 0.05, * <0.05, *** <0.001.

**Figure 7 molecules-25-04865-f007:**
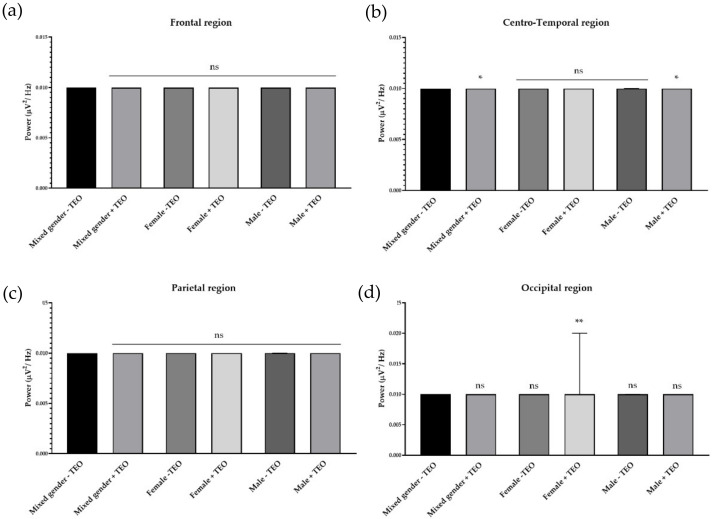
Effects of tangerine essential oil on high beta (25 Hz): (**a**) Frontal region; (**b**) Centro-Temporal region; (**c**) Parietal region; (**d**) Occipital region. *p* value: ns ≥ 0.05, * <0.05, ** <0.01.

**Figure 8 molecules-25-04865-f008:**
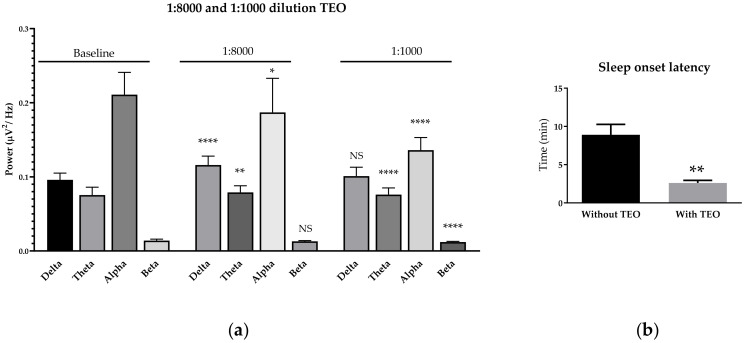
Effects of 1:8000 and 1:1000 oil dilution on brain waves (**a**). Sleep onset latency observed in female subjects (**b**). *p* value: ns ≥ 0.05, * <0.05, ** <0.01, **** <0.0001.

**Figure 9 molecules-25-04865-f009:**
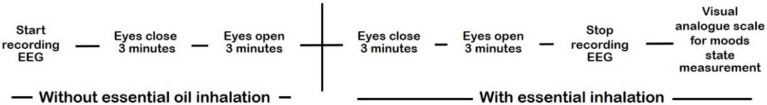
Experimental procedure divided into two trials. With tangerine essential oil inhalation and without tangerine essential oil inhalation.

**Table 1 molecules-25-04865-t001:** Chemical compositions of tangerine oil.

RT Min	% (Area)	Compound
5.17	1.05	α-Phellandrene
5.90	0.10	Camphene
6.76	1.36	β-Pinene
7.17	1.52	Sabinene
7.67	0.19	3-Carene
8.05	4.87	β-Myrcene
8.24	0.08	2,4,6-Octatriene, 3,4-dimethyl-
8.88	74.47	d-Limonene
9.49	10.86	γ-Terpinene
9.88	2.54	o-Cymene
10.55	0.19	Octanal
12.22	0.04	Nonanal
13.90	0.60	Linalool
Total	97.87	

**Table 2 molecules-25-04865-t002:** Representative brain wave power after inhaled tangerine essential oil.

Brainwaves	Electrodes	Without Essential Oil Inhalation (µV^2^)	With Essential Oil Inhalation (µV^2^)	*p* Value
Slow alpha wave	PZ	1.192	10.869	0.0397
Fast alpha wave	AF4	0.068	0.041	0.0015
Low beta wave	T7	0.028	0.041	0.0164

**Table 3 molecules-25-04865-t003:** Mean and SD of emotional state after in tangerine oil inhalation.

Emotions	Female		Male	
	Mean	SD	Mean	SD
Good	5.34	2.36	5.40	2.10
Active	4.96	1.01	4.59	1.79
Drowsy	4.63	2.75	3.68	3.22
Fresh	5.93	1.94	6.16	1.50
Romantic	1.54	1.46	2.49	1.88

**Table 4 molecules-25-04865-t004:** Baseline characteristics for brainwave studies. Data are shown as mean ± SD.

Gender	n	Age (Years)	Weight (kg)	Height (cm)	BMI (kg/m^2^)	Sleep (Hours)	Handedness
Female	10	21.6 ± 1.43	50.6 ± 5.78	161 ± 7.3	19.49 ± 2.49	7.12 ± 1.93	Right
Male	10	21 ± 0.67	59.9 ± 9.64	167 ± 6.34	21.44 ± 3.41	6.2 ± 2.2	Right

**Table 5 molecules-25-04865-t005:** Baseline characteristics for sleep onset latency study. Data are shown as mean ± SD.

Gender	n	Age (Years)	Weight (kg)	Height (cm)	BMI (kg/m^2^)	Handedness
Female	10	21.6 ± 0.66	53.1 ± 13	156.9 ± 3.28	21.56 ± 5.21	Right

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
