# Peer review of "Effects of Tangerine Essential Oil on Brain Waves, Moods, and Sleep Onset Latency"

_molecules, 2020, doi:10.3390/molecules25204865_

Round 1

Reviewer 1 Report

Comments and Suggestions for Authors

This is a well written manuscript describing the effect of tangerine peel essential oil on human brain waves power and sleep onset latency. This will be of interest to many in the field of natural products effects on human nervous system. The effect of different essential oils on brain e.g. lavender, jasmine or rose have been previously investigated  showing different results regarding to changes in brain waves’ power. These studies are citied by authors in presented paper in which the novelty are extensive data showing tangerine essential oil effects on brain response, sleep onset latency and emotions. While the results are extensively presented, there are a few points of revision:

1)The authors determined the content of compounds identified in tangerine oil by GC-MS. The results show that limonen is the most abundant component. Authors could make some efforts to discuss if this compound is responsible for observed effects or it is a synergistic effect of multiple ingredients of tangerine oil.

2) Also authors could attempts to explain observed gender difference between female and males responses to tangerine peel essential oil as changes in frequency of brain waves powers.

Minor comments:

The symbols a,b,c,d used for identification of four  brain regions are missing on four panels in the figures 3-7. However the names of brain regions appeared as titles of graphs presented on the figures. Thus, authors need to decide which type of brain region identification will use, since a,b,c,d symbols are mentioned in the descriptions of figures 3-7.

Author Response

Point 1: The authors determined the content of compounds identified in tangerine oil by GC-MS. The results show that limonene is the most abundant component. Authors could make some efforts to discuss if this compound is responsible for observed effects or it is a synergistic effect of multiple ingredients of tangerine oil.

Response 1: Thank you so much, to clarify, more discussions on limonene have been added into discussion part (line #170-177) according to the suggestion.

Point 2: Authors could attempt to explain observed gender difference between female and male responses to tangerine peel essential oil as changes in frequency of brain waves powers.

Response 2: The discussion about gender difference has been added into discussion part (line #222-237) according to the suggestion.

Point 3: The symbols a,b,c,d used for identification of four  brain regions are missing on four panels in the figures 3-7. However the names of brain regions appeared as titles of graphs presented on the figures. Thus, authors need to decide which type of brain region identification will use, since a,b,c,d symbols are mentioned in the descriptions of figures 3-7.

Response 3: The missing symbols a, b, c, d have been added in the figures. However, we offer both symbols and the name of brain region titles in the figure panels which may be easier for reader to understand the results.

Reviewer 2 Report

Title: Effects of Tangerine Essential Oil on Brain Waves, 3 Moods, and Sleep Onset Latency.

Supaya et al. studied the effect of Citrus tangerine, essential oil on human brain waves and sleep activity. The authors showed the impact of tangerine essential oil on the brain by measuring the electroencephalography (EEG) activity in 10 male and ten female subjects. This experimental study is exciting and has commercial value. However, the data based only on EEG would be interesting if authors studied some biochemical parameters like antioxidant enzymes and factors involved in sleep.

My Specific comments are,

  1. In the figure panel, label all subfigures, like 1a, 1b.
  2. What is the age and bodyweight of subjects that should be included in MS?
  3. What is the diet pattern of these subjects?
  4. What is the nature of work because this will impact a lot in sleep patterns?
  5. I strongly suggest the authors include the BMI of subjects used in this study. 

Author Response

First, we thank you for your kind review and suggestion. According to your suggestion, the biochemical parameters study will be further elucidated in our future work with full analysis of hypnogram.

For the specific comments

Point 1: In the figure panel, label all subfigures, like 1a, 1b.

Response 1: The symbols a, b, c, d have been added in the figures.

Point 2+5: What is the age and bodyweight and BMI of subjects that should be included in MS?

Response 2: The age, bodyweight, and BMI have been added into baseline characteristics table (table 4 and table 5)

Point 3: What is the diet pattern of these subjects?

Response 3:  the diet pattern of subjects has been mentioned in methodology as “The diet pattern of the subjects was the traditional Thai eating pattern” (line #260-261).

Supporting data: The traditional Thai eating pattern; in Thai meal, dishes are not served in courses. The various dishes are served at the same time and with rice, which is a main staple for the Thai population. Aquatic animals, meat, plants, spices, and herbs are used in preparing meals however large pieces of meat are rarely used in any dish. (Kosulwat, Vongsvat. (2002). The Nutrition and Health Transition in Thailand. Public health nutrition. 5. 183-9. 10.1079/PHN2001292.)

Point 4: What is the nature of work because this will impact a lot in sleep patterns?

Response 3:  All of the subject are undergraduate university students and has been mentioned in methodology (line #256)